# Comparative Analysis of Primary and Monovalent Booster SARS-CoV-2 Vaccination Coverage in Adults with and without HIV in Catalonia, Spain

**DOI:** 10.3390/vaccines12010044

**Published:** 2023-12-30

**Authors:** Daniel Kwakye Nomah, Juliana Reyes-Urueña, Lucía Alonso, Yesika Díaz, Sergio Moreno-Fornés, Jordi Aceiton, Andreu Bruguera, Raquel Martín-Iguacel, Arkaitz Imaz, Maria del Mar Gutierrez, Ramón W. Román, Paula Suanzes, Juan Ambrosioni, Jordi Casabona, Jose M. Miro, Josep M. Llibre

**Affiliations:** 1Center for Epidemiological Studies of Sexually Transmitted Diseases and HIV/AIDS in Catalonia (CEEISCAT), Department of Health, Government of Catalunya, 08916 Badalona, Spain; juliana.reyes81@gmail.com (J.R.-U.); lalonso@lluita.org (L.A.); ydiazr@iconcologia.net (Y.D.); smorenof@iconcologia.net (S.M.-F.); jaceiton@iconcologia.net (J.A.); abruguerar@iconcologia.net (A.B.); raquel@bisaurin.org (R.M.-I.); jcasabona@iconcologia.net (J.C.); 2Germans Trias i Pujol Research Institute (IGTP), 08916 Badalona, Spain; 3CIBER Epidemiologia y Salud Pública (CIBERESP), 08003 Barcelona, Spain; 4Departament de Pediatria, d’Obstetrícia i Ginecologia i de Medicina Preventiva i de Salut Publica, Universitat Autònoma de Barcelona, 08193 Bellaterra, Spain; 5Department of Infectious Diseases, Odense University Hospital, 5000 Odense, Denmark; 6Department of Infectious Diseases, Hospital Universitari de Bellvitge-(IDIBELL), 08907 L’Hospitalet de Llobregat, Spain; aimaz@bellvitgehospital.cat; 7Hospital de la Santa Creu i Sant Pau, 08041 Barcelona, Spain; mgutierrezma@santpau.cat; 8Agència de Qualitat i Avaluació Sanitàries de Catalunya, 08005 Barcelona, Spain; ramon.roman@gencat.cat; 9Hospital Universitari Vall d’Hebron, Vall d’Hebron Research Institute (VHIR), 08035 Barcelona, Spain; paula.suanzes@vhir.org; 10Hospital Clínic-Institut d’Investigacions Biomèdiques August Pi i Sunyer (IDIBAPS), University of Barcelona, 08036 Barcelona, Spain; ambrosioni@clinic.cat (J.A.); jmmiro@ub.edu (J.M.M.); 11CIBERINFEC, Instituto de Salud Carlos III, 28029 Madrid, Spain; 12Hospital Universitari Germans Trias i Pujol, 08916 Badalona, Spain; jmllibre@lluita.org

**Keywords:** HIV, SARS-CoV-2, COVID-19, vaccination, booster doses

## Abstract

People with HIV (PWH) may be more susceptible to SARS-CoV-2 infection and worse clinical outcomes. We investigated the disparity in SARS-CoV-2 vaccination coverage between PWH and those without HIV (PWoH) in Catalonia, Spain, assessing primary and monovalent booster vaccination coverage from December 2021 to July 2022. The vaccines administered were BNT162, ChAdOx1-S, mRNA-127, and Ad26.COV2.S. Using a 1:10 ratio of PWH to PWoH based on sex, age, and socioeconomic deprivation, the analysis included 201,630 individuals (183,300 PWoH and 18,330 PWH). Despite a higher prevalence of comorbidities, PWH exhibited lower rates of complete primary vaccination (78.2% vs. 81.8%, *p* < 0.001) but surpassed PWoH in booster coverage (68.5% vs. 63.1%, *p* < 0.001). Notably, complete vaccination rates were lower among PWH with CD4 <200 cells/μL, detectable HIV viremia, and migrants compared to PWoH (*p* < 0.001, all). However, PWH with CD4 < 200 cells/μL received more boosters (*p* < 0.001). In multivariable logistic regression analysis of the overall population, a prior SARS-CoV-2 diagnosis, HIV status, migrants, and mild-to-severe socioeconomic deprivation were associated with lower primary vaccination coverage, reflecting barriers to healthcare and vaccine access. However, booster vaccination was higher among PWH. Targeted interventions are needed to improve vaccine coverage and address hesitancy in vulnerable populations.

## 1. Introduction

Remarkable scientific and governmental investments have been made to develop multiple vaccine candidates against severe acute respiratory syndrome coronavirus 2 (SARS-CoV-2) in an unprecedented time [1]. These vaccines have proven to be an effective and viable approach to combat the ongoing pandemic and mitigate its socioeconomic and health impacts. The European Medicines Agency (EMA) has authorized eight vaccines for use in the European Union [2]. As of 22 November 2023, over 13.5 billion SARS-CoV-2 vaccine doses had been administered across 184 countries, making it the largest vaccination campaign in human history [3]. 

People with HIV (PWH) may be more susceptible to SARS-CoV-2 infection and face worse outcomes. According to a report from the World Health Organization (WHO), PWH have a 30% higher risk of mortality from COVID-19 after hospital admission compared to people without HIV (PWoH) [4]. Additionally, PWH may face poorer COVID-19 outcomes due to other social determinants of health, chronic comorbid conditions, and poor HIV control [5,6]. As a result, many countries prioritized PWH for vaccine eligibility. 

Existing evidence demonstrates that SARS-CoV-2 vaccines offer protection against COVID-19 by effectively reducing symptomatic infections and severe outcomes [7]. However, vaccine hesitancy among certain sub-populations [8,9] and the decline in IgG antibody levels after SARS-CoV-2 infection or vaccination [10] have hindered the full potential of vaccine protection. The emergence of SARS-CoV-2 variants with increased transmissibility and the ability to evade vaccine-induced immunity [11] is also a cause for concern. The uncertainties regarding the duration of vaccine protection and the impact of new variants underscored the importance of booster vaccinations [12]. Studies have shown that booster doses significantly decrease the risk of severe COVID-19 [13,14]. There are recommendations to administer booster doses to PWH with advanced immunosuppression or untreated HIV infection due to their increased risk of severe COVID-19 illness and potentially weaker immune response to SARS-CoV-2 vaccination [15]. In addition to their increased vulnerability, PWH may encounter barriers that limit their access to the crucial SARS-CoV-2 vaccinations [16]. 

Research on vaccination coverage among PWH is limited and lacks comparison with a matched sample from the general population [17,18]. Since vaccination strategies in many countries prioritize the public based on factors such as the nature of their jobs, age, presence of comorbidities, and other risk factors for adverse COVID-19 outcomes, matched studies are essential to assess the equity and effectiveness of current vaccination strategies, identify under-vaccinated groups, and provide valuable insights for future pandemics. The objective of this report is to compare primary and booster monovalent vaccination coverage among PWH with a well-matched representative sample of PWoH in Catalonia, Spain, and to identify subpopulations with low vaccination uptake to inform public health policies on ongoing vaccination strategies and future vaccination campaigns. 

## 2. Materials and Methods

### 2.1. Study Design and Population

We conducted a retrospective cohort study using data from the prospective PISCIS cohort linked with integrated healthcare, clinical, and surveillance registries through the Data Analysis Program for Research and Innovation in Health (PADRIS) [19] to obtain information on vaccination. PISCIS is an ongoing, population-based, longitudinal, systematic, prospective, and multicentre HIV cohort study of individuals receiving care in Catalonia and the Balearic Islands, Spain. Details of the cohort have been described elsewhere [20]. For the purposes of this study, we used participants receiving care in the 16 PISCIS hospitals in Catalonia, representing approximately 84% of all PLWH in the region.

People with HIV were matched 1:10 to HIV-negative individuals from the general population in Catalonia for sex at birth, 5-year age group, and socioeconomic deprivation using exact matching. The socioeconomic index is generated by the Catalan government based on the basic health area of residence (ABS, abbreviation in Catalan) to determine the socioeconomic levels of Catalonia residents [21] and takes into account the following indicators: the proportion of manual workers, the proportion of residents with low education levels, the proportion with low incomes, the rate of premature mortality, and the rate of avoidable hospitalization [21]. 

We excluded PWH who were not alive as of 27 December 2020, the day the vaccination campaign began in Spain, as well as those not in active clinical follow-up (those who have not used healthcare services for at least 12 months) to ensure the accurate estimation of vaccine coverage. HIV-negative individuals were classified as such if there was no record of HIV infection based on the absence of HIV International Classification of Diseases (ICD) codes. The study period was from 27 December 2020 to 19 July 2022.

### 2.2. Outcomes

We defined complete primary vaccination according to the criteria set by the WHO: (a) two doses of the BNT162 (Pfizer), mRNA-1273 (Moderna), or ChAdOx1-S (Oxford/AstraZeneca) vaccines; or (b) a single dose of Janssen Ad26.COV2.S [22]. Incomplete vaccination was defined as receiving only a single dose of the BNT162 (Pfizer), mRNA-1273 (Moderna), or ChAdOx1-S (Oxford/AstraZeneca) vaccines. Booster vaccinations were defined as any additional doses administered after completing the primary vaccination series [22]. 

### 2.3. Covariates

The sociodemographic covariates included age as of 1 January 2021, sex assigned at birth, country of origin classified as Spanish or non-Spanish, socioeconomic deprivation grouped into least deprivation, mild deprivation, or moderate-to-severe deprivation. The COVID-19-associated variables included are: history of SARS-CoV-2 diagnosis defined as a positive nucleic acid amplification test (NAAT) and/or antigen detection from respiratory samples [23]. Comorbidity covariates included the most prevalent conditions in the PWH population cohort: chronic respiratory disease, cardiovascular disease, chronic kidney or liver disease, neuropsychiatric conditions, diabetes, cancer, hypertension, obesity, and autoimmune disease. Comorbidity data were extracted using the ICD-10 codes (Appendix B). Among PWH, additional data were collected on years since HIV diagnosis, HIV transmission risk group (people who inject drugs [PWID], men who have sex with men [MSM], male heterosexual, female sexual transmission, and others), antiretroviral therapy (ART) reception, most recent CD4 cell count (categorized as <200 cells/μL, 200–499 cells/μL, and ≥500 cells/μL), and HIV plasma RNA viral load categorized as detectable and undetectable (<50 copies/mL).

### 2.4. Statistical Analysis 

We described the distribution of sociodemographic and clinical variables between PWH and PWoH to determine differences between the two populations. We used multivariable logistic regression models to assess the factors associated with complete vaccine reception and booster vaccinations, providing adjusted odds ratios (aOR) with 95% confidence intervals (95% CIs). The models were adjusted for age, sex, country of origin, socioeconomic deprivation, prior SARS-CoV-2 diagnosis, number of comorbidities, and HIV status. We calculated the cumulative incidence of complete vaccine reception and booster doses using Kaplan–Meier techniques from January 2021 to April 2022. We stratified the vaccine coverage analysis by HIV status, and among PWH, by country of origin (Spanish and non-Spanish), CD4 cell count categories, and HIV plasma RNA viral load. Log-rank tests were calculated to estimate the differences in cumulative vaccination coverage. We conducted subgroup analysis to investigate the factors associated with complete vaccine reception and booster vaccinations in both groups (Appendix A). We performed all analyses with R version 4.1.3 (R Project for Statistical Computing). A 2-sided *p*-value of <0.05 was considered statistically significant. 

### 2.5. Ethics

The Institutional Review Board of Germans Trias i Pujol Hospital in Badalona, Spain approved the PISCIS cohort study. Patient-level information obtained from PADRIS was anonymized and deidentified before analysis. This study followed the Strengthening the Reporting of Observational studies in Epidemiology (STROBE) guidelines.

## 3. Results

### 3.1. Baseline Characteristics of Study Population

A total of 18,330 PWH were matched to 183,300 PWoH in a ratio of 1:10. There were no differences in sex, age, and socioeconomic deprivation between the two groups. However, significant differences were observed regarding country of origin (*p* < 0.001), number of comorbidities (*p* < 0.001), and previous SARS-CoV-2 diagnosis (*p* < 0.001) between the two groups (Table 1).

Among the PWH, 15,062 individuals (82.2%) were male, and the majority (81.1%) fell within the age range of 31–60 years. The most common HIV acquisition risk group was MSM, accounting for 53.3%. The median (interquartile range [IQR]) CD4 cell count was 680 (486–908) cells/μL, with 627 PWH (3.4%) having a CD4 cell count below 200 cells/μL. The median (IQR) CD4/CD8 ratio was 0.85 (0.57–1.2), and 14,404 PWH (78.6%) had undetectable HIV RNA viremia (Table 1).

### 3.2. Vaccination Coverage 

Among the 201,630 individuals included in the study, 81.4% had received complete primary vaccination, while 63.5% had received booster doses. People with HIV had lower rates of complete vaccination compared to those without HIV (78.2% vs. 81.8%, *p* < 0.001). However, PWH had higher coverage of booster doses compared to the non-HIV group (68.5% vs. 63.1%, *p* < 0.001). The median duration in months between the primary vaccination series and the reception of a booster was similar in both groups at 6.4 months (IQR 6.0–7.1). Regarding the types of vaccines administered, the majority of study participants received the BNT162 BioNTech/Pfizer vaccine for the primary vaccination series (61.5%). However, for booster doses, the mRNA-1273 Moderna vaccine was more commonly administered (86.5%). The general HIV-negative population was more frequently vaccinated with BNT162 BioNTech/Pfizer; however, PWH vaccinated at hospitals received the mRNA-1273 Moderna as their primary dose (Table 2).

### 3.3. Factors Associated with Vaccine Coverage

In the overall population, a multivariable logistic regression analysis, adjusted for all potential confounders, revealed that PWH were less likely to receive the complete primary vaccine compared to PWoH (aOR 0.86; 95% CI 0.82–0.89). Other factors associated with lower odds of receiving the complete primary vaccine included non-Spanish origin (aOR 0.39; 95% CI 0.38–0.40), mild socioeconomic deprivation (aOR 0.87; 95% CI 0.84–0.90), moderate-to-severe socioeconomic deprivation (aOR 0.87; 95% CI 0.85–0.90), and a previous SARS-CoV-2 diagnosis (aOR 0.20; 95% CI 0.19–0.20) (Table 3).

Regarding booster vaccination, similar associations were observed. Non-Spanish origin (aOR 0.75; 95% CI 0.73–0.77), mild socioeconomic deprivation (aOR 0.80; 95% CI 0.78–0.83), moderate-to-severe socioeconomic deprivation (aOR 0.77; 95% CI 0.75–0.79), and a previous SARS-CoV-2 diagnosis (aOR 0.24; 95% CI 0.23–0.25) were all associated with lower odds of receiving booster monovalent vaccines. Increasing age was associated with increasing odds of receiving boosters (Table 3).

### 3.4. Comparing Primary Complete Vaccination and Boosters between Key HIV Groups and the HIV-Negative Population

Compared to PWoH, individuals living with HIV had higher vaccination rates against SARS-CoV-2 in the first 200 days of the vaccination campaign. However, after 16 months, complete vaccination coverage was significantly lower among PWH (*p* < 0.001). We observed a similar vaccination coverage between PWoH and PWH with CD4 counts >500 cells/μL. However, among PWH with CD4 counts <200 cells/μL, complete primary vaccination coverage was significantly lower (*p* < 0.001). Similarly, primary vaccination coverage was similar between PWoH and PWH with undetectable HIV viral loads but was significantly lower among PWH with detectable viral loads (*p* < 0.001). Significant differences were also observed in primary vaccination coverage between the host Spanish population and individuals of non-Spanish origin (*p* < 0.001) (Figure 1).

Regarding booster vaccinations, the coverage was higher among PWH compared to PWoH (*p* < 0.001). PWH with CD4 < 200 cells/μL received more boosters compared to PWoH (*p* < 0.001). Booster reception among PWH with undetectable viral loads; however, remained significantly lower compared to PWoH and PWH with undetectable HIV viral loads. In terms of country of origin, booster reception was lower among individuals of non-Spanish origin (Figure 2). 

## 4. Discussion

PWH may be more susceptible to severe COVID-19 outcomes [4]. Therefore, ensuring equitable access to SARS-CoV-2 vaccines for this vulnerable population is vital [24]. Furthermore, specific sub-populations, such as older individuals, those with lower CD4 cell counts, detectable HIV viremia, and chronic comorbidities, face elevated risks and potentially worse clinical outcomes from HIV/COVID-19 co-infection [6]. Researchers recommend that prevention strategies should target these particular sub-groups [6,25]. 

In our study, the primary vaccination coverage in the overall population (PWH and PWoH) was 81.4%, surpassing the regional average of 75.1% reported by the European Centre for Disease Prevention and Control (ECDC) [26]. Similarly, the observed coverage of booster vaccinations in our cohort was 65.3%, also exceeding the reported regional average of 54.8% [26]. These findings underscore the significant efforts made by the Government of Spain and the Spanish Agency for Medicines and Healthcare Products (AEMPS) to implement nationwide vaccination strategies, particularly emphasizing access for the most vulnerable groups. They also underline historical vaccine acceptance and favourable willingness to be vaccinated in Spain [27]. However, it is worth noting that while our observed primary vaccination rate exceeds the regional average, it falls slightly below the reported 84.9% for the same period in Spain [28]. This suggests that further work is needed to achieve optimal vaccination coverage in the general population of Catalonia.

The observed lower SARS-CoV-2 primary vaccination rates among PWH compared to PWoH in Catalonia are concerning. This trend aligns with a global HIV cohort [17] and a study conducted in New York [29], indicating a consistent pattern of reduced vaccination coverage among PWH. The observed disparities in SARS-CoV-2 vaccination rates among PWH could be attributed to various factors, including potential barriers and hesitancy toward vaccination [18]. Even before the pandemic, vaccine hesitancy was recognized as a significant global health concern by the World Health Organization [30]. Concerns about the safety of the new SARS-CoV-2 vaccines have been a primary reason for vaccine refusal, as highlighted in reports [31]. Furthermore, access barriers, including limited availability or insufficient information tailored to the needs of PWH, could contribute to lower vaccination rates within this population [31]. 

Multiple unmeasured factors, including differences in the nature of jobs, might influence this reduced vaccination rate. The study also identified sociodemographic factors such as migration status and socio-economic deprivation as predictors of lower vaccine uptake, mirroring the findings of a New York-based report [29]. Despite the Catalan Healthcare system offering universal and cost-free access to all citizens, regardless of administrative status, studies in Catalonia have shown a correlation between migrants, socio-economic deprivation, and limited healthcare service utilization [32]. Additionally, non-Spanish origin encompasses a wide range of factors including cultural disparities, language barriers, and specific community beliefs and practices which might significantly influence perceptions and access to vaccination services. Of particular concern is the lower coverage of complete primary vaccination among PWH with CD4 counts below 200 cells/μL and those with detectable HIV viral load, which mirrors findings from previous US studies [17,33]. These individuals are likely to be at higher risk of severe COVID-19 clinical outcomes due to their compromised immune status [6]. The presence of detectable viral loads has been associated with a younger age, a higher likelihood of missing medical appointments, and a lack of treatment adherence [34]. These factors could also partially explain why this important sub-population is under-vaccinated and underscores the necessity for comprehensive, patient-centered approaches to support PWH in achieving optimal health outcomes. Historically, PWH have shown hesitancy toward vaccinations compared to their HIV-negative counterparts, and understanding this reluctance in future studies could be crucial to tailoring effective interventions.

Consistent with an earlier study conducted in Catalonia [35], individuals previously infected with SARS-CoV-2 showed lower vaccine uptake. This can be linked to Catalonia’s vaccination strategy, which delayed schedules for individuals with prior infections for their subsequent vaccination until six months after a confirmed SARS-CoV-2 diagnosis [36], presuming some level of immunity from their past exposure. Evaluating both natural and vaccine-induced immunity is crucial in understanding COVID-19 risk, especially in high-transmission risk settings.

Recommendations for vaccinating PWH favored mRNA vaccines over Ad5 vector SARS-CoV-2 ones [37] due to concerns arising from the Step [38] and Phambili [39] studies, revealing increased HIV-1 acquisition risk in Ad5-vaccinated men. In our study, 75.8% of PWH initially received mRNA vaccines. This could result from using public spaces for vaccinations in the early phase of the pandemic to improve accessibility and coverage without compulsory HIV-status disclosure. However, during booster doses, 99.6% of PWH received mRNA vaccines, likely because booster vaccinations were handled by HIV units and vaccines without adenoviral vectors were prioritized. 

Despite the lower overall primary vaccination coverage, PWH showed higher rates of booster dose uptake compared to PWoH. This finding suggests that PWH and their healthcare providers may proactively seek additional doses to bolster their immune response, particularly following reports indicating inadequate immunogenicity and severe clinical outcomes from HIV/SARS-CoV-2 co-infection. It could also imply reluctance among the general population to receive booster shots, as reported in other settings [40,41]. Studies in the general population have linked this reluctance to perceived or reported side effects from the primary vaccination series, perceived (in)effectiveness of booster doses, low perception of COVID-19 risk, safety concerns, and lower education levels [40,41]. 

The higher odds of booster dose reception among PWH with CD4 levels below 200 cells/μL, aligning with public health recommendations [15], is encouraging due to their increased susceptibility to severe COVID-19 outcomes. However, the lower booster coverage among PWH with detectable viral loads requires attention. PWH with detectable viral loads might face challenges such as non-adherence to antiretrovirals, missing appointments, or engaging in risky behaviors, all of which could predict lower reception of booster doses. During the surge of the Omicron variant in Catalonia, the monovalent SARS-CoV-2 vaccine was introduced. Studies have demonstrated higher immunogenicity of monovalent booster doses against Omicron compared to a two-dose regimen [42,43]. Developing strategies to enhance booster vaccinations in vulnerable populations, like PWH, during this period was pertinent. These individuals stand to benefit greatly from booster doses due to their increased risk of breakthrough infections [44].

Our study has some notable strengths. To our knowledge, this is the first comprehensive evaluation of SARS-CoV-2 booster vaccination coverage among matched people with and without HIV. Additionally, the study used adequate matching of key sociodemographic factors to address potential differences between PWH and PWoH, enhancing the validity. 

The study has some limitations as well. Firstly, the socioeconomic deprivation measure is an ecological variable based on an individual’s place of residence. A person’s place of residence may indeed not necessarily reflect their socioeconomic deprivation. Secondly, we did not report data regarding the post-vaccination experiences of participants, particularly side effects from the primary vaccination doses, which might influence participants’ willingness to receive booster vaccinations. Thirdly, due to the nature of our study design, there might be residual confounding as certain variables, such as religion and occupation, factors that could impact vaccine reception, are not reported in our databases. Additionally, self-made home SARS-CoV-2 antigen test results are not available in our database. We are not able to report if this influenced vaccine reception. 

## 5. Conclusions

In conclusion, the study highlighted concerning discrepancies in SARS-CoV-2 vaccination rates between PWH and those without HIV in Catalonia, Spain. We observed lower primary vaccination rates among PWH compared to PWoH, even though PWH tended to have a higher prevalence of comorbidities. This indicates potential barriers to vaccination access or healthcare linkage among this group, especially for migrants, individuals experiencing socioeconomic deprivation, those with lower CD4 counts, and detectable HIV viral loads. However, the study uncovered a contrasting trend concerning booster vaccinations. PWH, particularly those with lower CD4 counts, were more likely to receive booster doses compared to PWoH. This is a positive finding suggesting increased awareness of booster shots among treating HIV physicians and PWH, particularly those with immunosuppression. Overall, the study underscores the need for targeted interventions to address the disparities in vaccination coverage among vulnerable populations. Improving access to primary vaccinations for PWH, especially those with lower CD4 counts and detectable viral loads, is crucial. Additionally, efforts to ensure equitable access to vaccines among migrants and socioeconomically deprived individuals are imperative. Improving HIV vaccination requires the involvement of HIV physicians, units, and effective communication emphasizing vaccine safety and efficacy.

## Figures and Tables

**Figure 1 vaccines-12-00044-f001:**
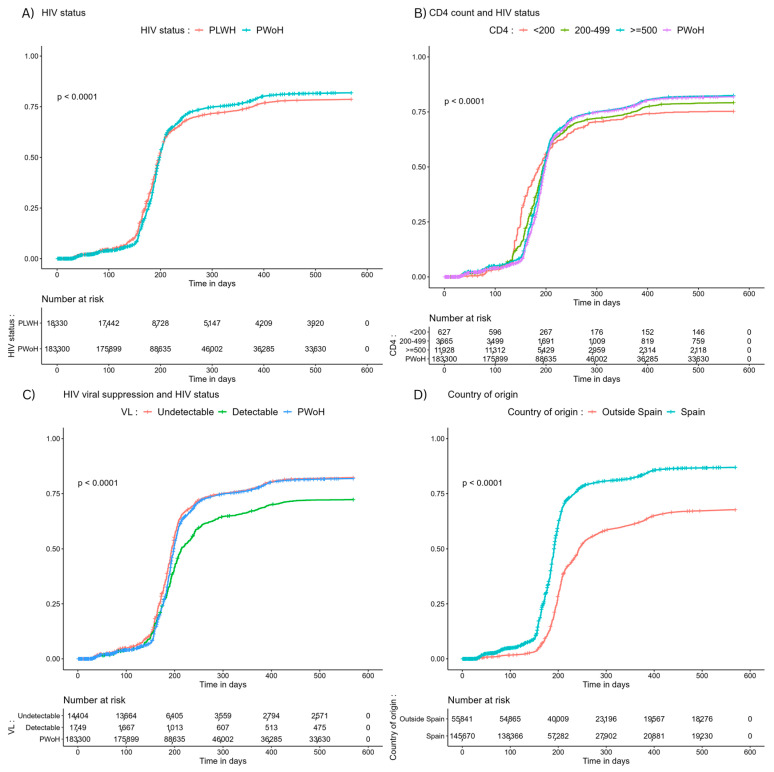
Cumulative incidence of complete primary SARS-CoV-2 vaccination stratified by (**A**) HIV status, (**B**) CD4 count and HIV status, (**C**) HIV viral suppression and HIV status, and (**D**) country of origin and HIV status. Abbreviations: PWH, people with HIV; PWoH, people without HIV; VL, HIV RNA viral load.

**Figure 2 vaccines-12-00044-f002:**
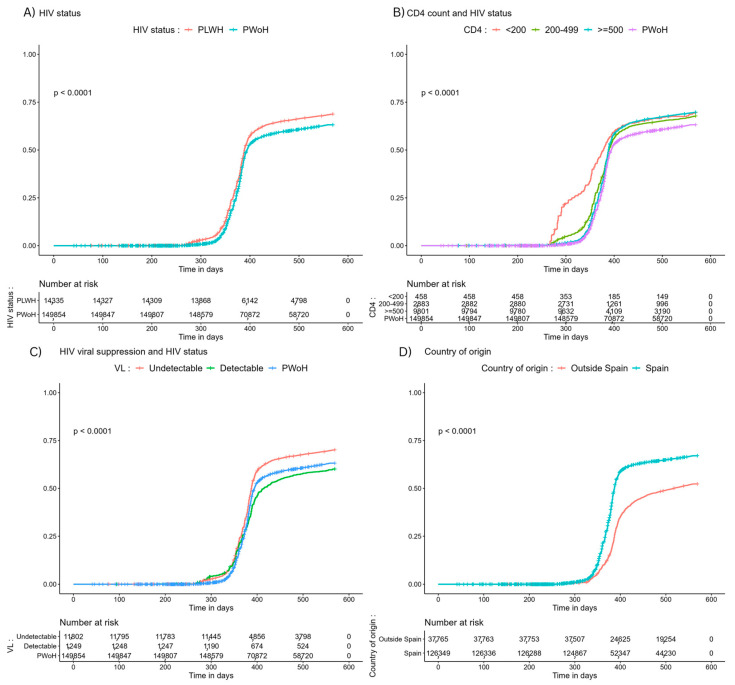
Cumulative incidence of booster SARS-CoV-2 vaccinations stratified by (**A**) HIV status, (**B**) CD4 count and HIV status, (**C**) HIV viral suppression and HIV status, and (**D**) country of origin and HIV status. Abbreviations: PWH, people with HIV; PWoH, people without HIV; VL, HIV RNA viral load.

**Table 1 vaccines-12-00044-t001:** Baseline characteristics of study participants according to HIV status.

	Total, n = 201,630	PWH, n = 18,330	PwoH, n = 183,300	*p*-Value
Characteristic	n (%)	n (%)	n (%)	
Sex ^a^				>0.99
Male	165,682 (82.2)	15,062 (82.2)	150,620 (82.2)	
Female	35,948 (17.8)	3268 (17.8)	32,680 (17.8)	
Age category, y ^b^				>0.99
16–30	17,985 (8.9)	1635 (8.9)	16,350 (8.9)	
31–40	48,081 (23.8)	4371 (23.8)	43,710 (23.8)	
41–50	62,909 (31.2)	5719 (31.2)	57,190 (31.2)	
51–60	52,602 (26.1)	4782 (26.1)	47,820 (26.1)	
61–70	15,004 (7.4)	1364 (7.4)	13,640 (7.4)	
>70	5049 (2.5)	459 (2.5)	4590 (2.5)	
Country of origin ^c^				<0.001
Spain	145,670 (72.2)	10,666 (58.2)	135,004 (73.7)	
Outside Spain	55,841 (27.7)	7662 (41.8)	48,179 (26.3)	
Missing	119 (0.1)	2 (0)	117 (0.1)	
Socioeconomic deprivation *				>0.99
Least deprived	99,836 (49.5)	9076 (49.5)	90,760 (49.5)	
Mildly deprived	38,544 (19.1)	3504 (19.1)	35,040 (19.1)	
Moderately/severely deprived	58,652 (29.1)	5332 (29.1)	53,320 (29.1)	
Missing	4598 (2.3)	418 (2.3)	4180 (2.3)	
Number of comorbidities				<0.001
0	87,154 (43.2)	5024 (27.4)	82,130 (44.8)	
1	46,042 (22.8)	4083 (22.3)	41,959 (22.9)	
2	29,102 (14.4)	3299 (18)	25,803 (14.1)	
3	18,348 (9.1)	2387 (13)	15,961 (8.7)	
≥4	20,984 (10.4)	3537 (19.3)	17,447 (9.5)	
Type of comorbidities				
Respiratory disease	18,472 (9.2)	3852 (21)	14,620 (8)	<0.001
Cardiovascular disease	20,974 (10.4)	2925 (16)	18,049 (9.8)	<0.001
Autoimmune disease	17,120 (8.5)	2019 (11)	15,101 (8.2)	<0.001
Chronic kidney disease	11,360 (5.6)	1622 (8.8)	9738 (5.3)	<0.001
Chronic liver disease	6764 (3.4)	3530 (19.3)	3234 (1.8)	<0.001
Neuropsychiatric conditions	59,822 (29.7)	9107 (49.7)	50,715 (27.7)	<0.001
Diabetes (type I and II)	10,949 (5.4)	1043 (5.7)	9906 (5.4)	0.1
Metabolic disease	40,981 (20.3)	4221 (23)	36,760 (20.1)	<0.001
Cancer	9004 (4.5)	1821 (9.9)	7183 (3.9)	<0.001
Hypertension	36,362 (18)	3688 (20.1)	32,674 (17.8)	<0.001
Obesity	28,874 (14.3)	1801 (9.8)	27,073 (14.8)	<0.001
Previous SARS-CoV-2 diagnosis				<0.001
Yes	25,093 (12.4)	2454 (13.4)	22,639 (12.4)	
No	176,537 (87.6)	15,876 (86.6)	160,661 (87.6)	
HIV acquisition risk group				
PWID		2360 (12.9)		
MSM		9761 (53.3)		
Male heterosexual		2443 (13.3)		
Female sexual tranmission		2419 (13.2)		
Other		519 (2.8)		
Missing		828 (4.5)		
Years since HIV diagnosis, median (IQR)		11.57 (5.91–18.57)		
CD4 count (cells/μL) category				
<200		627 (3.4)		
200–499		3665 (20)		
≥500		11,928 (65.1)		
Missing		2110 (11.5)		
CD4 count (cells/μL), median (IQR)		680 (486–908)		
CD4/CD8 ratio, median (IQR)		0.85 (0.57–1.2)		
Plasma HIV-RNA				
Detectable		1749 (9.5)		
Undetectable		14,404 (78.6)		
Missing		2177 (11.9)		
Years on ART, median (IQR) ^d^		8.75 (4.16–14.41)		
On Treatment				
Yes		14,685 (80.1)		
No		3645 (19.9)		

Abbreviations: PWH, people with HIV; PWoH, people without HIV; SARS-CoV-2, severe acute respiratory syndrome coronavirus 2; IQR, interquartile range; PWID, people who inject drugs; MSM, men who have sex with men; ART, antiretroviral therapy. ^a^ Sex as assigned birth. ^b^ Age for all patients was as of 1 January 2021. ^c^ Country of origin was as indicated by the Public Data Analysis for Health Research and Innovation Program of Catalonia (PADRIS), recorded as Spanish or Non-Spanish. ^d^ Years on ART was defined as the difference in time between the first treatment administration date to the latest treatment date or the latest hospital visit if the last treatment date is missing. * Socioeconomic deprivation is based on the index of the Catalan government according to the basic health area (ABS) of residence. This index is based on five indicators which are: proportion of manual workers, proportion of residents with low education level, proportion with low income, rate of premature mortality, and rate of avoidable hospitalization.

**Table 2 vaccines-12-00044-t002:** SARS-CoV-2 vaccination coverage between people with and without HIV.

	Total	PWH	PWoH	*p*-Value
Primary vaccination	n (%)	n (%)	n (%)	<0.001
Unvaccinated	29,606 (14.7)	3343 (18.2)	26,263 (14.3)	
Incomplete	7835 (3.9)	652 (3.6)	7183 (3.9)	
Complete	164,189 (81.4)	14,335 (78.2)	149,854 (81.8)	
Primary vaccination type				<0.001
BNT162	105,743 (61.5)	7924 (52.9)	97,819 (62.3)	
ChAdOx1-S	16,668 (9.7)	1436 (9.6)	15,232 (9.7)	
mRNA-1273	29,505 (17.2)	3737 (24.9)	25,768 (16.4)	
Ad26.COV2.S	13,346 (7.8)	1321 (8.8)	12,025 (7.7)	
Combined	6762 (3.9)	569 (3.8)	6193 (3.9)	
Booster doses				<0.001
Yes	104,332 (63.5)	9823 (68.5)	94,509 (63.1)	
No	59,857 (36.5)	4512 (31.5)	55,345 (36.9)	
Booster doses type				<0.001
BNT162	13,973 (13.4)	1413 (14.4)	12,560 (13.3)	
ChAdOx1-S	26 (0)	4 (0)	22 (0)	
mRNA-1273	90,250 (86.5)	8372 (85.2)	81,878 (86.6)	
Ad26.COV2.S	10 (0)	2 (0)	8 (0)	
Combined	40 (0)	17 (0.2)	23 (0)	
Other	33 (0)	15 (0.2)	18 (0)	
Median time between primary and booster dose, months (IQR)	6.44 (5.98–7.1)	6.44 (5.92–7.13)	6.44 (6.02–7.1)	<0.001

Abbreviations: PWH, people with HIV; PWoH, people without HIV; IQR, interquaartile range.

**Table 3 vaccines-12-00044-t003:** Factors associated with (a) complete and (b) booster vaccine reception in logistic regression analysis.

	Complete Primary Vaccination		Booster Vaccination	
	aOR (95% CI)	*p*-Value	aOR (95% CI)	*p*-Value
**HIV Status**				
Negative	1 (ref)		1 (ref)	
Positive	0.86 (0.82, 0.89)	<0.001	1.41 (1.36, 1.47)	<0.001
**Sex**				
Male	1 (ref)		1 (ref)	
Female	1.1 (1.07, 1.14)	<0.001	1.03 (1, 1.06)	0.091
**Age category, y**				
16–30	1 (ref)		1 (ref)	
31–40	1.3 (1.25, 1.36)	<0.001	1.61 (1.54, 1.68)	<0.001
41–50	1.79 (1.72, 1.87)	<0.001	2.76 (2.65, 2.88)	<0.001
51–60	2.21 (2.11, 2.31)	<0.001	4.67 (4.46, 4.89)	<0.001
61–70	2.12 (1.99, 2.27)	<0.001	9.88 (9.26, 10.55)	<0.001
>70	2.73 (2.42, 3.07)	<0.001	17.48 (15.53, 19.67)	<0.001
**Place of Birth**				
Spain	1 (ref)		1 (ref)	
Outside Spain	0.39 (0.38, 0.4)	<0.001	0.75 (0.73, 0.77)	<0.001
Missing				
**Socioeconomic deprivation ***				
Least deprived	1 (ref)		1 (ref)	
Mildly deprived	0.87 (0.84, 0.9)	<0.001	0.8 (0.78, 0.83)	<0.001
Moderately/severely deprived	0.87 (0.85, 0.9)	<0.001	0.77 (0.75, 0.79)	<0.001
**Number of comorbidities**				
0	1 (ref)		1 (ref)	
1	1.26 (1.22, 1.3)	<0.001	1.01 (0.98, 1.04)	0.599
2	1.45 (1.39, 1.51)	<0.001	1.07 (1.04, 1.11)	<0.001
3	1.64 (1.56, 1.73)	<0.001	1.13 (1.08, 1.18)	<0.001
≥4	1.78 (1.69, 1.88)	<0.001	1.27 (1.21, 1.32)	<0.001
**Previous SARS-CoV-2 diagnosis**				
No	1 (ref)		1 (ref)	
Yes	0.2 (0.19, 0.2)	<0.001	0.24 (0.23, 0.25)	<0.001

Abbreviations: OR, odds ratio; aOR, adjusted odds ratio; SARS-CoV-2, severe acute respiratory syndrome coronavirus 2. Models adjusted for age, sex, country of origin, socioeconomic deprivation, prior SARS-CoV-2 diagnosis, number of comorbidities, and HIV status. * Socioeconomic deprivation is based on the index of the Catalan government according to the basic health area (ABS) of residence. This index is based on five indicators which are: proportion of manual workers, proportion of residents with low education level, proportion with low income, rate of premature mortality, and rate of avoidable hospitalization.

## Data Availability

The study protocol and statical code are available from the corresponding author upon request. The data for this study accessed on 1 October 2022 are available at the Centre for Epidemiological Studies of Sexually Transmitted Diseases and HIV/AIDS in Catalonia (CEEISCAT), the coordinating centre of the PISCIS cohort, the PADRIS, and from each of the collaborating hospitals upon request via https://pisciscohort.org/contacte/.

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
