# Peer review of "Comparative Analysis of Primary and Monovalent Booster SARS-CoV-2 Vaccination Coverage in Adults with and without HIV in Catalonia, Spain"

_vaccines, 2023, doi:10.3390/vaccines12010044_

Round 1

Reviewer 1 Report

Comments and Suggestions for Authors

The manuscript addresses a crucial public health concern by thoroughly investigating SARS-CoV-2 vaccination rates among people with HIV (PWH) in Catalonia, Spain. By examining both primary and booster vaccination coverage, the study sheds light on notable disparities between PWH and those without HIV, emphasizing the need for targeted interventions. The findings, revealing lower primary vaccination rates among PWH despite a higher prevalence of comorbidities, underscore potential barriers to healthcare access within this vulnerable population. Conversely, the positive trend of increased booster doses among PWH, particularly those with lower CD4 counts, signifies heightened awareness and proactive efforts among treating HIV physicians and individuals with immunosuppression. The manuscript's significance lies in its contribution to the ongoing discourse on equitable vaccine distribution and healthcare access, emphasizing the importance of tailored interventions to bridge gaps in vaccination coverage among marginalized groups, ultimately informing public health strategies and interventions to mitigate the impact of COVID-19 on vulnerable populations. The writing style of the manuscript is generally clear and precise, effectively conveying complex information. The authors employ a scientific and formal tone, ensuring the credibility of their research. However, there are certain shortcomings which must be addressed.

1.     The abstract presents a comprehensive investigation into the SARS-CoV-2 vaccination coverage among people with HIV (PWH) compared to those without HIV (PWoH) in Catalonia, Spain. The study is commendable for employing a robust methodology, including a 1:10 ratio matching of PWH to PWoH based on relevant demographic factors. However, the abstract does not delve into the reasons behind the observed disparities in vaccination coverage among PWH, particularly those with CD4 <200 cells/μL, detectable HIV viremia, and migrants.

2.     Understanding the specific barriers and challenges faced by these subgroups could provide valuable insights for targeted interventions. Additionally, the abstract lacks information on the specific types of vaccines administered, which could impact the generalizability of the findings.

3.     While the abstract emphasizes the need for targeted interventions, it falls short in proposing concrete recommendations or strategies to address vaccine coverage disparities and hesitancy in vulnerable populations.

4.     Moreover, the abstract would benefit from a discussion of potential confounding factors and limitations related to the socioeconomic and healthcare access variables considered in the multivariable logistic regression analysis.

5.     The introduction provides a comprehensive overview of the global efforts in developing and administering SARS-CoV-2 vaccines, emphasizing the vulnerability of people with HIV (PWH) to severe outcomes and the importance of vaccination. However, the introduction lacks a clear articulation of the specific research gap or question that the study aims to address.

6.     While it highlights the general challenges related to vaccine hesitancy, declining antibody levels, and concerns about emerging variants, it does not explicitly state the need for a study comparing vaccination coverage between PWH and a matched sample from the general population. The introduction could benefit from a more focused and explicit statement of the research objective, outlining why this specific comparison in Catalonia, Spain, is crucial.

7.     Providing a clearer rationale and context for the study would strengthen the introduction and help readers better understand the significance of the research within the broader landscape of COVID-19 vaccination.

8.     The methodology section outlines a retrospective cohort study using data from the PISCIS cohort in Catalonia, Spain, to investigate SARS-CoV-2 vaccination coverage among people with HIV (PWH) compared to a matched sample of HIV-negative individuals (PWoH). While the study design and statistical analysis are generally well-described, However, the matching criteria, including sex, age, and socioeconomic deprivation, are mentioned, but potential confounding factors related to health behaviors, healthcare access, or other relevant variables are not thoroughly discussed.

9.     The methodology does not explicitly discuss the representativeness of the 16 collaborating hospitals in Catalonia, potentially limiting the generalizability of the findings.

10.  The results section provides a comprehensive overview of the study outcomes, including baseline characteristics, vaccination coverage, and factors associated with vaccine reception among people with HIV (PWH) and those without HIV (PWoH). However, several limitations warrant consideration. Overall, while the study contributes valuable insights, a thorough exploration of potential confounding factors would enhance the robustness and generalizability of the findings.

11.  The discussion section provides a thorough analysis of the vaccination coverage among people with HIV (PWH) and those without HIV (PWoH), shedding light on important disparities and trends. However, there are some notable limitations that warrant consideration. The discussion appropriately highlights the vulnerability of PWH to severe COVID-19 outcomes and emphasizes the importance of equitable vaccine access. Nevertheless, the interpretation of findings could be strengthened by delving deeper into the reasons behind the observed disparities, such as potential barriers to vaccination or hesitancy among PWH.

12.  The study acknowledges the lower primary vaccination rates among PWH, particularly those with compromised immune status, but a more nuanced exploration of the underlying factors influencing this discrepancy would enhance the discussion's depth.

13.  Additionally, the discussion could benefit from addressing the limitations of the study more explicitly, such as the ecological nature of the socioeconomic deprivation measure, the lack of data on post-vaccination experiences, and the potential impact of unreported variables like religion and occupation.

14.  Moreover, acknowledging the absence of information on self-made home SARS-CoV-2 antigen test results and its potential influence on vaccine reception would add transparency to the interpretation of the results.

15.  The conclusion section effectively summarizes the key findings of the study and emphasizes the concerning vaccination rate discrepancies between people with HIV (PWH) and those without HIV in Catalonia, Spain. The identification of potential barriers to vaccination access, especially among specific subgroups like migrants and individuals facing socioeconomic deprivation, is well-articulated. The positive trend of higher booster vaccination rates among PWH with lower CD4 counts is appropriately highlighted. However, the conclusion could benefit from a more explicit discussion of the specific interventions or strategies that could address the identified disparities.

16.  While the need for targeted interventions is emphasized, providing concrete recommendations based on the study's findings would enhance the practical implications of the research.

17.  Additionally, acknowledging the limitations discussed in the previous sections, such as the lack of in-depth exploration of underlying factors influencing vaccination rates and the ecological nature of the socioeconomic deprivation measure, would contribute to a more balanced and transparent conclusion.

Comments on the Quality of English Language

Minor editing is required.

Author Response

Reviewer 1

The manuscript addresses a crucial public health concern by thoroughly investigating SARS-CoV-2 vaccination rates among people with HIV (PWH) in Catalonia, Spain. By examining both primary and booster vaccination coverage, the study sheds light on notable disparities between PWH and those without HIV, emphasizing the need for targeted interventions. The findings, revealing lower primary vaccination rates among PWH despite a higher prevalence of comorbidities, underscore potential barriers to healthcare access within this vulnerable population. Conversely, the positive trend of increased booster doses among PWH, particularly those with lower CD4 counts, signifies heightened awareness and proactive efforts among treating HIV physicians and individuals with immunosuppression. The manuscript's significance lies in its contribution to the ongoing discourse on equitable vaccine distribution and healthcare access, emphasizing the importance of tailored interventions to bridge gaps in vaccination coverage among marginalized groups, ultimately informing public health strategies and interventions to mitigate the impact of COVID-19 on vulnerable populations. The writing style of the manuscript is generally clear and precise, effectively conveying complex information. The authors employ a scientific and formal tone, ensuring the credibility of their research. However, there are certain shortcomings which must be addressed.

  1. The abstract presents a comprehensive investigation into the SARS-CoV-2 vaccination coverage among people with HIV (PWH) compared to those without HIV (PWoH) in Catalonia, Spain. The study is commendable for employing a robust methodology, including a 1:10 ratio matching of PWH to PWoH based on relevant demographic factors. However, the abstract does not delve into the reasons behind the observed disparities in vaccination coverage among PWH, particularly those with CD4 <200 cells/μL, detectable HIV viremia, and migrants. Understanding the specific barriers and challenges faced by these subgroups could provide valuable insights for targeted interventions.

Response: Thank you for the comments. Thank you for your feedback. Due to the 200-word limit, the abstract could not cover reasons for vaccination disparities among PWH. We thoroughly discuss these points in the manuscript's discussion section.

  1. Additionally, the abstract lacks information on the specific types of vaccines administered, which could impact the generalizability of the findings.

Response: Thank you for your input. The abstract now includes details on the specific vaccines administered. Please refer to Line 40 for the information: Vaccines administered were BNT162, ChAdOx1-S, mRNA-127, and Ad26.COV2.S.

  1. While the abstract emphasizes the need for targeted interventions, it falls short in proposing concrete recommendations or strategies to address vaccine coverage disparities and hesitancy in vulnerable populations.

Response: Thank you for your feedback. We recognize the significance of proposing actionable strategies to curb the identified disparities. But again, we are limited by a 200-world limit in the abstract. Our manuscript covers recommended strategies, notably emphasizing the proactive role of HIV treating physicians and awareness initiatives. These recommendations are detailed in the discussion and conclusion sections of the manuscript.

In line 376: Overall, the study underscores the need for targeted interventions to address the disparities in vaccination coverage among vulnerable populations. Improving access to primary vaccinations for PWH, especially those with lower CD4 counts and detectable viral loads, is crucial. Additionally, efforts to ensure equitable access to vaccines among migrants and socioeconomically deprived individuals are imperative. Improving HIV vaccination requires involvement of HIV physicians, units, and effective communication emphasizing vaccine safety and efficacy.

  1. Moreover, the abstract would benefit from a discussion of potential confounding factors and limitations related to the socioeconomic and healthcare access variables considered in the multivariable logistic regression analysis.

Response: Thank you for highlighting the importance of addressing confounding factors and limitations related to socioeconomic and healthcare access variables. To mitigate these concerns, our methodology included rigorous measures such as a 1:10 ratio matching based on sex, age, and socioeconomic deprivation, as indicated in line 41 of the abstract. Additionally, while the abstract could not detail the adjustments in the models due to the word limit, the methods section explains these adjustments, including age, sex, country of origin, socioeconomic deprivation, prior SARS-CoV-2 diagnosis, comorbidities, and HIV status (line 151). These factors were selected based on established studies recognizing them as potential barriers to SARS-CoV-2 vaccination uptake.

  1. The introduction provides a comprehensive overview of the global efforts in developing and administering SARS-CoV-2 vaccines, emphasizing the vulnerability of people with HIV (PWH) to severe outcomes and the importance of vaccination. However, the introduction lacks a clear articulation of the specific research gap or question that the study aims to address.

Response: Thank you for your feedback. We believe the introduction addresses the research gap and objectives clearly in the final paragraph. Specifically, in line 85 of the introduction, we outlined the scarcity of comparative studies on vaccination coverage among PWH, emphasizing the necessity for matched studies to assess equity, effectiveness of vaccination campaigns, and identify under-vaccinated groups.

Line 85-94 reads: Research on vaccination coverage among PWH is limited and lack comparison with a matched sample from the general population [17,18]. Since vaccination strategies in many countries prioritize the public based on factors such as nature of jobs, age, presence of comorbidities, and other risk factors for adverse COVID-19 outcomes, matched studies are essential to assess the equity and effectiveness of current vaccination strategies, identify under-vaccinated groups, and provide valuable insights for future pandemics. The objective of this report is to compare primary and booster monovalent vaccination coverage among PWH with a well-matched representative sample of PWoH in Catalonia, Spain, and to identify subpopulations with low vaccination uptake to inform public health policies on ongoing vaccination strategies and future vaccination campaigns.

Previous large-scale studies conducted globally by Fulda ES et al. (Fulda, E.S. et al. COVID-19 Vaccination Rates in a Global HIV Cohort. J. Infect. Dis. 2022, 225, 603–607, doi:10.1093/infdis/jiab575) and a substantial study from China by Liu Y et al. (Liu, Y. et al.. COVID-19 Vaccination in People Living with HIV (PLWH) in China: A Cross Sectional Study of Vaccine Hesitancy, Safety, and Immunogenicity. Vaccines 2021, 9, 1458) both lacked well-matched groups for comparing vaccination coverage. This underscored the need for studies like our current one to comprehend vaccination inequities, understand barriers, and identify undervaccinated subgroups, especially considering that PWH are a potentially vulnerable population to COVID-19.

  1. While it highlights the general challenges related to vaccine hesitancy, declining antibody levels, and concerns about emerging variants, it does not explicitly state the need for a study comparing vaccination coverage between PWH and a matched sample from the general population. The introduction could benefit from a more focused and explicit statement of the research objective, outlining why this specific comparison in Catalonia, Spain, is crucial. Providing a clearer rationale and context for the study would strengthen the introduction and help readers better understand the significance of the research within the broader landscape of COVID-19 vaccination.

Response: Thank you Reviewer 1 for the comment. The objective of the study extends beyond generating knowledge pertinent solely to Catalonia; instead, it aims to contribute insights relevant across Europe and other high-income settings within the population of people living with HIV. The intent is to foster broader applicability and facilitate the extrapolation of findings to inform strategies, policies, and interventions addressing vaccination disparities among individuals with HIV.

As highlighted in the previous response, previous large-scale studies conducted globally by Fulda ES et al and a substantial study from China by Liu Y et al both lacked well-matched groups for comparing vaccination coverage. This underscored the need for studies like our current one to comprehend vaccination inequities, understand barriers, and identify undervaccinated subgroups, especially considering that PWH are a potentially vulnerable population to COVID-19.

  1. The methodology section outlines a retrospective cohort study using data from the PISCIS cohort in Catalonia, Spain, to investigate SARS-CoV-2 vaccination coverage among people with HIV (PWH) compared to a matched sample of HIV-negative individuals (PWoH). While the study design and statistical analysis are generally well-described, However, the matching criteria, including sex, age, and socioeconomic deprivation, are mentioned, but potential confounding factors related to health behaviors, healthcare access, or other relevant variables are not thoroughly discussed.

Response: Thank you, Reviewer 1, for recognizing the strengths of our methodology and providing valuable feedback for improvement.

In our methodology, socioeconomic deprivation was a crucial matching variable, comprising factors such as the proportion of manual workers, residents with low education levels, low-income populations, premature mortality rates, and avoidable hospitalization rates in residential areas, as detailed in line 109.

Line 109: The socioeconomic index is generated by the Catalan government based on the basic health area of residence (ABS, abbreviation in Catalan) to determine the socioeconomic levels of Catalonia residents [21] and takes into account the following indicators: proportion of manual workers, proportion of residents with low education level, proportion with low income, rate of premature mortality and rate of avoidable hospitalization [21].

Regarding potential confounding factors beyond our cohort's scope, we addressed these limitations in our discussion. Line 344 acknowledges the absence of data on post-vaccination experiences and line 347, the study's design limitations include potential residual confounding due to unreported variables like religion and occupation.

Line 344: Secondly, we did not report data regarding the post-vaccination experiences of partici-pants, particularly side effects from the primary vaccination doses, which might influence the willingness to receive booster vaccinations.

Line 347: Thirdly, due to the nature of our study design, there might be residual confounding as certain variables, such as religion and occupation, factors that could impact vaccine re-ception, are not reported in our databases.

  1. The methodology does not explicitly discuss the representativeness of the 16 collaborating hospitals in Catalonia, potentially limiting the generalizability of the findings.

Response: Thank you for highlighting the concern regarding the representativeness of the collaborating hospitals in Catalonia.

In our manuscript, we refer to the PISCIS cohort study (Bruguera, A.; Nomah, D.; et al. Cohort Profile: PISCIS, a Population-Based Cohort of People Living with HIV in Catalonia and Balearic Islands. Int. J. Epidemiol. 2023, dyad083) and clarify in line 104 that it encompasses approximately 84% of people living with HIV in the region. The 16% not covered primarily include individuals without administrative residence permits in Spain, as per current legislation, which restricts the use of their data for research purposes. Additionally, some patients might receive care in secondary-care hospitals not yet enrolled in the cohort.

Regarding generalizability, our matching approach on sex, age, and socioeconomic deprivation aimed to ensure close similarities between the two populations.

  1. The results section provides a comprehensive overview of the study outcomes, including baseline characteristics, vaccination coverage, and factors associated with vaccine reception among people with HIV (PWH) and those without HIV (PWoH). However, several limitations warrant consideration. Overall, while the study contributes valuable insights, a thorough exploration of potential confounding factors would enhance the robustness and generalizability of the findings.

Response: Thank you for your feedback. In our methodology, we detailed the strategies employed to address potential confounding factors. However, we acknowledge certain limitations regarding factors beyond our control, which are comprehensively discussed in the study's limitations section. We will be glad to consider any other specific confounding factors that have not yet been discussed in our manuscript.

  1. The discussion section provides a thorough analysis of the vaccination coverage among people with HIV (PWH) and those without HIV (PWoH), shedding light on important disparities and trends. However, there are some notable limitations that warrant consideration. The discussion appropriately highlights the vulnerability of PWH to severe COVID-19 outcomes and emphasizes the importance of equitable vaccine access. Nevertheless, the interpretation of findings could be strengthened by delving deeper into the reasons behind the observed disparities, such as potential barriers to vaccination or hesitancy among PWH.

Response: Thank you for acknowledging the strengths of our study and recognizing the thorough analysis of vaccination coverage among PWH and those without HIV. We have taken your feedback seriously and have delved deeper into the discussion about the reasons behind the observed disparities and hesitancy, particularly addressing potential barriers to vaccination and highlighting concerns about safety, access limitations, and tailored information for PWH. We hope these additions further enhance the interpretation of our findings.

Line 285-292: The observed disparities in SARS-CoV-2 vaccination rates among PWH could be attributed to various factors, including potential barriers and hesitancy toward vaccination [18]. Even before the pandemic, vaccine hesitancy was recognized as a significant global health concern by the World Health Organization [30]. Concerns about the safety of the new SARS-CoV-2 vaccines have been a primary reason for vaccine refusal, as highlighted in reports [31]. Furthermore, access barriers, including limited availability or insufficient in-formation tailored to the needs of PWH, could contribute to lower vaccination rates within this population [31].

  1. The study acknowledges the lower primary vaccination rates among PWH, particularly those with compromised immune status, but a more nuanced exploration of the underlying factors influencing this discrepancy would enhance the discussion's depth.

Response: Thank you for your feedback. We appreciate your insight into the need for a more nuanced exploration of the factors influencing the lower primary vaccination rates among PWH, especially those with compromised immune status. We have taken this suggestion into consideration and have expanded upon the discussion, specifically delving into the complexities associated with detectable viral loads among PWH.
Line 306: The presence of detectable viral loads has been associated with a younger age, a higher likelihood of missing medical appointments, and lack of treatment adherence [34]. These factors could also partially explain why this important sub-population are undervaccinated and underscores the necessity for comprehensive, patient-centered approaches to support PWH in achieving optimal health outcomes.

  1. Additionally, the discussion could benefit from addressing the limitations of the study more explicitly, such as the ecological nature of the socioeconomic deprivation measure, the lack of data on post-vaccination experiences, and the potential impact of unreported variables like religion and occupation.

Response: Thank you. We admitted the limitation of the socioeconomic variable being ecological as the first limitation of our study.
Line 355: The study has some limitations as well. Firstly, the socioeconomic deprivation measure is an ecological variable based on an individual’s place of residence. A person's place of residence may indeed not necessarily reflect their socioeconomic deprivation.

Under the methods, we also explained how the Government of Catalonia calculates the socioeconomic variable.
Line 109: The socioeconomic index is generated by the Catalan government based on the basic health area of residence (ABS, abbreviation in Catalan) to determine the socioeconomic levels of Catalonia residents [21] and takes into account the following indicators: propor-tion of manual workers, proportion of residents with low education level, proportion with low income, rate of premature mortality and rate of avoidable hospitalization [21].

  1. Moreover, acknowledging the absence of information on self-made home SARS-CoV-2 antigen test results and its potential influence on vaccine reception would add transparency to the interpretation of the results.

Response: Thank you. Even though the main goal of the study is to access vaccination coverage in the two populations, we indicated in line 363: Additionally, self-made home SARS-CoV-2 antigen test results are not available our database. We are not able to account if this influenced vaccine reception.  

  1. The conclusion section effectively summarizes the key findings of the study and emphasizes the concerning vaccination rate discrepancies between people with HIV (PWH) and those without HIV in Catalonia, Spain. The identification of potential barriers to vaccination access, especially among specific subgroups like migrants and individuals facing socioeconomic deprivation, is well-articulated. The positive trend of higher booster vaccination rates among PWH with lower CD4 counts is appropriately highlighted. However, the conclusion could benefit from a more explicit discussion of the specific interventions or strategies that could address the identified disparities.

Response: Thank you for your valuable feedback. We have addressed this concern within the discussion section and in line with your suggestion, we have included a recommendation in the conclusion emphasizing the involvement of HIV physicians, units, and effective communication to improve vaccination rates among PWH.

Line 379: Improving HIV vaccination requires involvement of HIV physicians, units, and effective communication emphasizing vaccine safety and efficacy.

  1. While the need for targeted interventions is emphasized, providing concrete recommendations based on the study's findings would enhance the practical implications of the research.

Response: Thank you and we believe this is similar to the previous comment. In the conclusion, we provided information on implications and recommendations of this current study. Line 375: Overall, the study underscores the need for targeted interventions to address the disparities in vaccination coverage among vulnerable populations. Improving access to primary vaccinations for PWH, especially those with lower CD4 counts and detectable viral loads, is crucial. Additionally, efforts to ensure equitable access to vaccines among migrants and socioeconomically deprived individuals are imperative. Improving HIV vaccination re-quires involvement of HIV physicians, units, and effective communication emphasizing vaccine safety and efficacy.

  1. Additionally, acknowledging the limitations discussed in the previous sections, such as the lack of in-depth exploration of underlying factors influencing vaccination rates and the ecological nature of the socioeconomic deprivation measure, would contribute to a more balanced and transparent conclusion.

Response: Thank you for highlighting the need to address the study's limitations. In lines 354-363, we discussed these limitations, including the ecological nature of the socioeconomic deprivation measure and other factors such as the absence of post-vaccination experiences and potential confounding variables.

Lines 354-363: The study has some limitations as well. Firstly, the socioeconomic deprivation measure is an ecological variable based on an individual’s place of residence. A person's place of residence may indeed not necessarily reflect their socioeconomic deprivation. Secondly, we did not report data regarding the post-vaccination experiences of participants, particularly side effects from the primary vaccination doses, which might influence the willingness to receive booster vaccinations. Thirdly, due to the nature of our study design, there might be residual confounding as certain variables, such as religion and occupation, factors that could impact vaccine reception, are not reported in our databases. Addition-ally, self-made home SARS-CoV-2 antigen test results are not available our database. We are not able to account if this influenced vaccine reception.  

Reviewer 2 Report

Comments and Suggestions for Authors

I appreciate the scientific quality and the public health relevance of your publication.

Almost no comments or suggestions for improvements.

As a limitation that you mentionned, the deprivation factor linked with job and employement may influence the willingness for complete vaccination among PWH.( reference to the New York study)

In the recommendation section to improve access to vaccination and booster, the engagement of HIV specific units for PWH seems to be a positive lesson from your observations and discussions.

Last , the vaccine coverage has been identified in this publication as the key outcome indicator. From a public health point of view , the reduction of morbidity and mortality due to more effective vaccination coverage is the targetted outcome.

Author Response

Reviewer 2

I appreciate the scientific quality and the public health relevance of your publication.

Almost no comments or suggestions for improvements.

  1. As a limitation that you mentionned, the deprivation factor linked with job and employement may influence the willingness for complete vaccination among PWH.( reference to the New York study)

Response: Thank you Reviewer 2. We have acknowledged in our study's limitations section that the deprivation factor associated with employment and job-related aspects could potentially impact the inclination for vaccination among PWH. This consideration aligns with insights from a referenced New York study, underscoring the significance of these socioeconomic factors in influencing vaccination willingness among this population.

Line 354-363: The study has some limitations as well. Firstly, the socioeconomic deprivation measure is an ecological variable based on an individual’s place of residence. A person's place of residence may indeed not necessarily reflect their socioeconomic deprivation. Secondly, we did not report data regarding the post-vaccination experiences of participants, particularly side effects from the primary vaccination doses, which might influence the willingness to receive booster vaccinations. Thirdly, due to the nature of our study design, there might be residual confounding as certain variables, such as religion and occupation, factors that could impact vaccine reception, are not reported in our databases. Addition-ally, self-made home SARS-CoV-2 antigen test results are not available our database. We are not able to account if this influenced vaccine reception.  

  1. In the recommendation section to improve access to vaccination and booster, the engagement of HIV specific units for PWH seems to be a positive lesson from your observations and discussions.

Response: Thank you. We have included this information in the conclusions of the study adding it as a recommendation from the study.

Line 379: Improving HIV vaccination requires involvement of HIV physicians, units, and effective communication emphasizing vaccine safety and efficacy.

  1. Last, the vaccine coverage has been identified in this publication as the key outcome indicator. From a public health point of view , the reduction of morbidity and mortality due to more effective vaccination coverage is the targetted outcome.

Response: Thank you for your insight. Our study's aim is to compare vaccination rates among specific populations rather than assess vaccination impact on morbidity and mortality. We focus on informing vaccination strategies for future public health policies in Catalonia, Spain.

Round 2

Reviewer 1 Report

Comments and Suggestions for Authors

The authors have addressed the comments and suggestions and have sufficiently improved the manuscript. Now, I am confident to recommend it for publication in its present form.